# The Socio-Educational Adaptation of Secondary School Migrant Students in Sicily: Migrant Generation, School Linguistic Mediation and Teacher Proactivity Factors

**José Juan Carrión-Martínez** [1], **Stefania Ciaccio** [2], **María del Mar Fernández-Martínez** [3,*], **Carmen María Hernández-Garre** [1] and **María Dolores Pérez-Esteban** [1]

1 Department of Education, University of Almería, 04120 Almeria, Spain; jcarrion@ual.es (J.J.C.-M.); cmhgarre@ual.es (C.M.H.-G.); mpe242@ual.es (M.D.P.-E.)
2 IPSSAR Pietro Piazza, 92019 Sciacca, Italy; stefania.ciaccio@ipsseoapiazza.it
3 Department of Sociology, Social Work and Public Health, University of Huelva, 21007 Huelva, Spain
* Correspondence: mar.fernandez@dstso.uhu.es

**Abstract:** This study aims to analyze the implications of linguistic mediation processes and educational proactivity in schools for the socio-educational adaptation of immigrant students. The study is based on empirical research and the perspectives of the main actors: the immigrant students themselves. To this end, a non-experimental and descriptive quantitative methodology was used. The sample consisted of 100 students of foreign origin enrolled in an Italian school located in a typical socio-cultural environment. The results show significant differences in linguistic mediation and socio-educational variables and differences in expectations of progress and social adaptation of students born outside Italy vis-a-vis students who, although born in Italy, are still considered foreigners. It will also analyze teaching proactivity as a communication facilitator for first-generation immigrant pupils born outside Italy. In conclusion, it is noted that, to favor multicultural environments where all students, regardless of their origin, feel accepted, integrated, and welcomed, it is necessary to utilize all available resources to promote improvements in teaching-learning processes and strengthen social relations.

**Keywords:** immigrant students; socio-educational adaptation; linguistic mediation; teacher proactivity; secondary education

## 1. Introduction

The integration of ethnic minorities within Western countries is undoubtedly one of the most critical problems for these nations, mainly due to the relentless increase in migratory flows in recent years (Arango et al. 2018; Gatta 2019; Paredes García 2020; Tuzi 2019). According to Ares Mateos (2020), the social integration of these immigrants is a complex process in which education is unanimously accepted as one of the most useful strategies for integrating immigrants into society (Adams-Ojugbele and Mashiya 2020; Dennison 2021).

Various studies (Alonso-Bello et al. 2020; Maskileyson et al. 2021; Santana et al. 2018) argue that social and labor integration are distinctive processes since their success is conditioned mainly by the level of education attained by the immigrants. However, the lack of social cohesion and the difficulty of young immigrants integrating into the host country often cause conflict and violence (Naz 2020).

Studies such as those developed by Ravecca (2010) show that the link between the level of education of second-generation migrants and their degree of integration is not always absolute. The degree of adaptation to new contexts is measured by the level of education achieved and their academic results. Similar results were found in the research developed by Morales et al. (2021), in which the self-perceived skills in mathematics and sciences of immigrant students were linked to integration and social relationships.

According to González Rábago (2014), the social context in which the different integration processes occur and the atmosphere that provokes cognitive, physical, or emotional development (Adams-Ojugbele and Mashiya 2020) undoubtedly determine the fate of future generations. It is not surprising that segments of the population in which young people are integrated depend heavily on their success or failure (Lacroix et al. 2020).

The educational field is conditioned by a network of multidimensional relations in which both socioeconomic factors (Gillen-O'Neel et al. 2021) and family (Mir 2019) are very relevant when it comes to alleviating educational inequalities. As stated by González Rábago (2014), it is vitally important to bear in mind the society of origin and its cultural aspects as these immigrants maintain strong ties to their countries of origin.

One of the most pervasive problems in classrooms with a large presence of immigrant students (Ester Sánchez 2016) is the difficulty in carrying out various curricular activities, mainly due to the lack of unified language proficiency (Salem-Gervais and Raynaud 2020). Linguistic and cultural mediation is understood as a process aimed at facilitating communication between two or more speakers who fail to communicate directly due to linguistic or cultural barriers (Lopes 2020; Pérez-Vázquez 2010; Trovato 2014). Thus, models such as the CCMEn (Antón-Solanas et al. 2020) help teachers to promote teaching-learning processes of cultural competence within multicultural environments; thus, all linguistic mediations carried out within the classroom are perceived as ideal tools for meeting the needs of students and developing the teaching activity of educators (Phipps 2013). Mediation is grounded in the different communicative activities that together with production, interaction, and reception favor the teaching-learning processes of any foreign language (Stramkale 2018), with educational centers being ideal places in which to put into practice the various cultural and linguistic tools that promote coexistence (Maurissen et al. 2020) and knowledge of other cultures (Guichot-Reina 2021). Schools are also environments that enable eradicating the seeds of racism and segregation (Carpani et al. 2011; Salem-Gervais and Raynaud 2020).

The current educational context is complex. It must enroll and educate immigrant students with or without language proficiency and also adapt the curriculum contents and didactic strategies to the needs of these students (Babane 2020; Cavicchiolo et al. 2020) so that they acquire essential instrumental learning and social skills that facilitate the ability to develop in the new environment (Soriano Ayala et al. 2019). This complicated situation, coupled with the lack of willingness of teachers (Murua-Carton et al. 2012) and their limited training to deal with new multicultural contexts (Rizova et al. 2020), makes it difficult to act in these new multicultural contexts (Rizova et al. 2020), or improve the integration of this group (Jolliffe and Speers 2016; Li 2018).

Nowadays, classrooms in European schools are places where inclusive values and acceptance of differences must converge (Gogolin 2021). One of the critical pieces for schools to reflect society lies in proactive teaching. As mediators, teachers can anticipate their students' needs (Olivier et al. 2021) and design active intervention strategies to alleviate differences (Carvalho et al. 2020), thus motivating students to learn. They can also provide challenges that give rise to different ways of learning, especially in the case of students of foreign origin (Heacox 2012; Marriott Toledo and Zambrano García 2018).

As Santos Loor et al. (2019) demonstrate in their study, educational spaces are environments that faithfully reflect society and the ever rapid pace of change. These new scenarios should make students feel welcome, and offer students comfort, safety, dignity, acceptance and integration within the group (Adams-Ojugbele and Mashiya 2020; Salem-Gervais and Raynaud 2020). The latter two aspects, feeling accepted and integrated within the group, represent a turning point when referring to first- or second-generation immigrant students (Maskileyson et al. 2021). First-generation immigrants are born in a country different from their new country of residence and hold foreign citizenship (Bonifazi and Paparusso 2019). Second-generation immigrants, although encompassing a wide range of specific circumstances, are often grouped under the same nomenclature and include unaccompanied minors, children reunited with their parents in the host country, children of mixed couples

(an immigrant and an autochthonous citizen), and all children of immigrants born in the host country (Maskileyson et al. 2021; Salvatori 2006).

For this conglomerate of people, first- and second-generation immigrants, acceptance within a group and feeling integrated can present additional difficulties for the simple fact that they are of diverse origin (Rutland et al. 2012). According to García-Bacete et al. (2010) and Thorjussen (2020), social acceptance is defined as the degree to which individuals feel loved, accepted, or rejected within their group of equals. It is a very significant variable in the study of socio-educational adaptation.

Being integrated into different contexts (educational, social, and family), having friends, and feeling accepted are evolutionary milestones that make up optimal cognitive, social and emotional development (Monjas 2010).

In line with the scenario mentioned above, the research questions that have guided this research are:

- What is the perception that non-EU foreign students have about language mediation and didactic facilitation at school? Does the gender or educational level of parents influence the socio-educational adaptation of immigrant pupils?
- Do linguistic mediations and teaching proactivity that occur in school favor these students' integration?
- Are there differences in socio-educational adaptation between first- and second-generation immigrant students?

In this research, the use of the term first- and second-generation immigrants are limited to operational categories, eschewing epistemological debates that are often far removed from the teaching-learning process. This study does not participate in other analyses on this much-debated issue (Checa and Arjona 2009), nor on the existence of possible intermediate generations (Arcarons and Muñoz-Comet 2018; Checa and Arjona 2009; Kern et al. 2020; Portes and Aparicio 2013). We define first-generation immigrant students as those born outside Italy and second-generation immigrant students as those born in Italy.

## 2. Methodology

For this research, a non-experimental and descriptive quantitative methodology was used to address research questions (McMillan and Schumacher 2012).

### 2.1. Objectives

- Describe the perception about linguistic mediation and didactic facilitation at school that non-EU foreign students have.
- Study the relationships between gender and educational level of parents with socio-educational adaptation.
- Analyse the relationship between the mediations that occur in the school, both linguistic and of teaching proactivity, and educational and social acceptance.
- Identify whether the expectation of socio-educational acceptance of immigrant students is based on belonging to the first or second generation

### 2.2. Participants

This study is based on the responses from a sample of 100 students enrolled in secondary education at the FFP Instituto in Italy. The selection of students was intentional as they were directly involved with this research and thus deemed to be the most suitable candidates for obtaining relevant information. Despite not being an extensive sample that will condition us to the type of inferential statistical treatment, we consider it a success to have 100 non-EU students studying in a single Italian school. Regarding the descriptions of the sample, the variables involved are reflected in Table 1.

**Table 1.** Descriptive characteristics of the sample.

| Data | | Frequency | Percentages |
|---|---|---|---|
| Gender | Male | 34 | 34.0 |
| | Female | 66 | 66.0 |
| Age | Minor | 54 | 54.0 |
| | Adult | 46 | 46.0 |
| Siblings | No | 9 | 9.0 |
| | Yes | 90 | 90.0 |
| Origin | Asian | 30 | 30.0 |
| | African | 18 | 18.0 |
| | Latin American | 52 | 52.0 |
| Has lived elsewhere other than Italy | No | 77 | 77.0 |
| | Yes | 23 | 23.0 |
| Have you benefited from linguistic facilitation interventions? | No | 80 | 80.0 |
| | Yes | 20 | 20.0 |
| Does the school make these linguistic facilitations available? | No | 31 | 31.0 |
| | Yes | 69 | 69.0 |

Source: Own compilation.

The selection of the secondary school is based on the characteristics of the population in the area since it is in the historic center of the city in which many foreigners of Moroccan, Tunisian, and Chinese origin live. The school reflects these demographics. Although native-Italian students are also enrolled in the school, the high proportion of first-generation immigrant students, including some unaccompanied foreign minors and second-generation immigrant students, makes it an ideal context for this study.

### 2.3. Instruments

A validated and well-developed instrument has been used in this study. The questionnaire's validity has been obtained through an expert validation procedure and the consistency matrix (Argentero 2000; Barroso Osuna and Cabero Almenara 2013; De Liaño and Pascual-Ezama 2012). A closed questionnaire is used ad hoc to narrow the range of responses and simplify the subsequent analysis (Hernández Sampieri et al. 2014). This consists of 58 closed items in which dichotomous questions are combined with Likert scales with four response levels. From the 58 primary items, 2 of them are used as independent variables (20 and 21) and another 11 are used to construct a macrovariable (grouping in terms of the arithmetic mean of several primary items). To this microvariable we must add the 7 initial items on sociodemographic variables.

### 2.4. Procedure and Analysis of Information

Regarding the analysis of the data, descriptive and frequency statistics were calculated to establish a preliminary analysis of sociodemographic variables (gender, education of the mother and father, mediation and proactivity (linguistic mediation, pedagogical mediation) and socio-educational adaption (perception of acceptance, school satisfaction, expectation of progress). It was followed by an analysis of the possible relationships between sociodemographic variables and socio-educational adaptation variables. Given that the robustness analysis of the sample does not meet the normality and homoscedasticity criteria required for the application of parametric tests, non-parametric tests have been applied (Figure 1).

| Research questions | | Statistical tests |
|---|---|---|
| What is the perception of non-EU foreign students about language mediation and didactic facilitation at school? | | Descriptives Binomial test |
| Does the gender or educational level of parents influence the socio-educational adaptation of immigrant pupils? | gender | Mann-Whitney U Test |
| | education level of parents | Kruskal-Wallis Test |
| Do linguistic mediations and between teaching proactivity that occur in school favor these students' integration? | | Mann-Whitney U Test Kruskal-Wallis Test |
| Are there differences in socio-educational adaptation between first- and second-generation immigrant students? | | Mann-Whitney U Test Pearson´s Chi-square |

**Figure 1.** Relationship questions and statistical tests.

## 3. Results

### 3.1. Description of Sociodemographic Variables, Perception of Mediation, and Proactivity in the School

The results associated with the sociodemographic variables show a vast difference between the number of male and female students who make up the sample since there are more female participants (66%) than male (34%).

Regarding the parents' level of education, it was observed that mothers had either an intermediate or higher level of education, with 43% possessing an intermediate level of education compared to 33% with a higher level of education.

Furthermore, the results related to the educational level of the fathers achieved identical results. Those with intermediate or higher education also predominate among fathers, although the difference between intermediate (39%) and higher (36%) is somewhat lower than in mothers.

In contrast, the mediation and proactivity variables' results show that 80% of students have not perceived any linguistic facilitation in the school compared to 20% who claimed to have received it (Table 2).

**Table 2.** Binomial test on linguistic facilitation.

| | | Category | N | Proportion Observed | Proportion of Sample | Significance (Bilateral) |
|---|---|---|---|---|---|---|
| Have you benefited from linguistic facilitation interventions? | Group 1 | no | 80 | 0.80 | 0.50 | 0.000 |
| | Group 2 | yes | 20 | 0.20 | - | |
| | Total | | 100 | 1.00 | - | |

A very high level of significance is attained through a binomial comparison evincing that such a distribution reflects that students lack appreciation or perception of this linguistic aid or facilitation (Table 2).

While the school makes these linguistic facilitations available to students, as confirmed by 69% of respondents, the high statistical significance attained compared to those who do not recognize what the school offers (Table 3) is evidence of a contradiction or conscious renunciation of such aid.

Regarding the linguistic facilitations used by teachers, there is a non-significant difference between the 53% of the students who claim that these facilitations have been employed, compared with 47% of respondents who claim they have not.

**Table 3.** Binomial test on the provision of facilitations.

| | Category | N | Proportion Observed | Proportion of Sample | Significance (Bilateral) |
|---|---|---|---|---|---|
| Does the school make these linguistic facilitations available? | Group 1 | yes | 69 | 0.69 | 0.50 | 0.000 |
| | Group 2 | no | 31 | 0.31 | - | |
| | Total | | 100 | 1.00 | - | |

This implies disharmony since, on the one hand, there is a significant prevalence of students who do not perceive or have not used linguistic or mediation aids, although, in the main, they recognize their existence in the school. However, the respondents are very divided when assessing the proactivity of teachers in this issue, although, in this respect, there is no significant differential distribution.

*3.2. Intragroup Acceptance, Global Satisfaction, Expectation of Progress and Socio-Educational Adaptation and Their Relationships with Gender, as Well as Educational Level of Their Father and Mother*

Concerning the basic descriptions of the variables: Intragroup Acceptance, Global Satisfaction, Expectation of Progress, and Socio-educational Adaptation of immigrant students, the results reveal a significantly higher value than expected in a normal distribution. This leads to a significant increase in normal distribution, leading us to interpret that students perceive themselves as enjoying a high level of intragroup acceptance, global satisfaction, expectation of progress, and socio-educational adaptation (Table 4).

**Table 4.** Kolmogorov-Smirnov normality test of a sample.

| | | Intragroup Acceptance | Overall Satisfaction | Expectation of Educational Progress | Socio-Educational Adaptation |
|---|---|---|---|---|---|
| N | | 100 | 100 | 100 | 100 |
| Normal parameters | Average | 1.6317 | 1.6500 | 1.6133 | 1.7930 |
| | Dev. | 0.49367 | 0.72648 | 0.70547 | 0.37044 |
| Sample Data | | 0.092 | 0.185 | 0.168 | 0.132 |
| Sig. asymptotic (bilateral) | | 0.037 * | 0.000 * | 0.000 * | 0.000 * |

* Correction of the significance of Lilliefors.

Having presented the basic description of the variables, a comparative analysis was carried out between the sociodemographic variables and the socio-educational adaptation variables using the non-parametric tests of Mann-Whitney and Kruskal-Wallis.

3.2.1. Relationship with the Gender Variable

Following the application of Mann-Whitney's U test to compare group ranges, it is observed that although males show greater overall satisfaction, expectation of educational progress, and socio-educational adaptation, female students have a greater degree of intragroup acceptance. These differences, however, are not significant.

3.2.2. Relationship to the Education Level of Parents

Concerning the educational level attained by the respondents' parents, in both cases, both the mother's and father's educational level, the non-parametric Kruskal-Wallis test has been applied. In this instance, no statistically significant differences between these variables and the level of education attained by the mother or the father were found.

*3.3. Relationships between Linguistic Mediation with Intragroup Acceptance, Global Satisfaction, Expectation of Progress and Socio-Educational Adaptation of Immigrant Students*

The results obtained in the comparative variables between linguistic mediation and socio-educational adaptation with the Mann-Whitney U test show that, in terms of linguistic

interventions and facilitation, these are significantly related to both intragroup acceptance and socio-educational adaptation (Table 5).

**Table 5.** Mann-Whitney U Test data on language facilitation interventions.

|  | Intragroup Acceptance | Overall Satisfaction | Expectation of Educational Progress | Socio-Educational Adaptation [a] |
|---|---|---|---|---|
| Mann-Whitney U Test | 516.500 | 728.000 | 710.500 | 538.500 |
| Sig. asymptotic (bilateral) | 0.014 | 0.523 | 0.432 | 0.024 |

[a] Grouping variable: Have you benefited from linguistic facilitation interventions?

### 3.4. Relationships between Linguistic Mediation Resources with Intragroup Acceptance, Global Satisfaction, Expectation of Progress and Socio-Educational Adaptation of Immigrant Students

Regarding the use of the school's resources, and always from the students' perspectives, the results show no statistically significant differences.

### 3.5. Relationships between Teaching Proactivity with Intragroup Acceptance, Global Satisfaction, Expectation of Progress and Socio-Educational Adaption of Immigrant Students

Teaching proactivity also stands out, showing insignificant results in the Mann-Whitney U test in all the groups that make up the variables: Intragroup Acceptance, Global Satisfaction, Expectation of Progress, and Socio-educational Adaptation.

### 3.6. Relationships between Intragroup Acceptance, Global Satisfaction, Expectation of Progress and Socio-Educational Adaptation of Immigrant Students and Their Categorization as First- or Second-Generation Immigrants

Concerning the relationships between immigrant student variables and whether they are categorized as first- or second-generation immigrants, Table 6 shows disparate statistically significant situations. While there are no significant differences in intragroup acceptance or overall satisfaction, there are differences in expectations of perceived socio-educational progress and adaptation, presenting more favorable values in the case of first-generation (born outside Italy) immigrant students.

**Table 6.** Statistical data on first- and second-generation immigrants.

|  | Intragroup Acceptance | Overall Satisfaction | Expectation of Educational Progress | Socio-Educational Adaptation |
|---|---|---|---|---|
| Mann-Whitney U Test | 1056.000 | 1188.500 | 865.500 | 903.000 |
| Sig. asymptotic (bilateral) | 0.187 | 0.685 | 0.007 | 0.017 |

Grouping variable: First/Second Generation.

### 3.7. Relationships between Language Facilitation, Facilitated Teaching Resources and Teacher Proactivity and Their Categorization as First- or Second-Generation Immigrants

Finally, with regard to the relationship between linguistic facilitation, teaching resources made available by the school and teacher proactivity and first-or second-generation immigrant students, while there are no significant differences in the perception of linguistic facilitation and resources of the school between the two groups (born in Italy/versus born abroad), there is a significant difference in the perception of the teacher's proactivity in making language communication easier. In this case, as seen in Table 7, we find that the first-generation group, even those born abroad, perceives a higher level of teacher performance in a significantly differentiated way.

**Table 7.** Chi-square tests.

|  | Value | Significance |
|---|---|---|
| Pearson's Chi-square | 8.101 | 0.004 |
| Num. of valid cases | 100 | |

Zero cells (0.0%) have an expected count of less than 5. The minimum expected count is 22.09.

## 4. Discussion

Schools are ideal places for young people to acquire social skills (Jolliffe and Speers 2016) and develop academically (Salem-Gervais and Raynaud 2020), through relationships with adults (Olivier et al. 2021), with their peers (Adams-Ojugbele and Mashiya 2020) and with the materials and resources available in their local environment.

Currently, schools are obliged to care for students from diverse cultural and linguistic backgrounds (Babane 2020), which combined with, in some cases, a lack of motivation (Murua-Carton et al. 2012) and inadequate teacher training (Rizova et al. 2020), causes inclusive education to become an increasingly utopian dream.

The most important contributions of this work are related to the link between linguistic mediation and teaching proactivity as facilitators of the socio-educational adaptation of immigrant students, which is its principal objective.

Regarding the first specific objective of this study, the data shows the relationship between the variables of gender, the education of the father and mother, and socio-educational adaptation. The tests carried out show that there are no statistically significant relationships regarding the sex of the participants or with the level of education of both parents. This is in line with the studies by Alivernini et al. (2019) and Vitoroulis and Vaillancourt (2015), showing that although non-acceptance within the group is related to immigration, its relation to gender is unclear (MacMullin et al. 2020). Cavicchiolo et al. (2020) found that the group, individual, or family characteristics are not as relevant as language proficiency for promoting immigrant children's inclusion in school.

The non-significance of differences between male and female students concerning the socio-educational acceptance variable contrasts with those obtained by Cavicchiolo et al. (2020), whose study shows that female students are twice as vulnerable in class and that this type of acceptance has an impact on their friendship group.

Concerning the educational level attained by both parents, it is observed that these are not significantly related to the socio-educational adaptation of their children. Similarly, the study by (Rizova et al. 2020) observed that parents with little school education combine all their efforts to achieve quality education for their children because an education in which different linguistic or social origins are actively working together will be more inclusive (Zilka 2020).

However, the data reflect that the sample perceived social adaption as "sufficient". This is in line with the results by Senra (2010) in which very satisfactory scores were obtained about the social skills of immigrant students, and those of Adams-Ojugbele and Mashiya (2020) in which their respondents, young immigrants, stated that thanks to the social interactions they developed in their context, mainly in school, they had a positive feeling about the socio-emotional perceptions of belonging and acceptance.

The second objective, the relationship between the perception of linguistic mediation in the school context or teaching proactivity with educational and social acceptance, shows that linguistic interventions and facilitation significantly and positively affect intragroup acceptance and socio-educational adaptation of immigrant students. These results are in line with those achieved by Larrañaga et al. (2020) in their research on the adaptation of immigrants to the context of bilingualism. Their study stated that immigrant students immersed in bilingualism or monolingualism achieve a high level of school adaptation. They emphasized that environments prone to developing bilingualism achieve higher social adaptation levels and linguistic identity, thus favoring positive intergroup relationships. Likewise, the study carried out by Cavicchiolo et al. (2020) states that the social inclusion

of immigrant students is subject to the learning of the national language even in second-generation immigrant pupils, showing that linguistic facilitation for language acquisition will significantly aid these inclusive processes and can be developed successfully.

The non-significance of teacher proactivity concerning the adaptive profile variables under study contrasts with those by Babane (2020), in which it is shown how teachers use different strategies to support and facilitate the learning of the new language by their immigrant students. It was observed that teaching proactivity and the mediations employed facilitate healthy relationships with adults and their peers and positively impact comprehensive development.

An example of teacher proactivity as one of the fundamental pillars for encouraging students to participate in teaching-learning processes is reflected in the development of a teaching-learning model of cultural competence within a multicultural context (CCMen) (Antón-Solanas et al. 2020) in which educational processes are adapted to the needs of students for the learning of a second language. Thus, Antón-Solanas et al. (2020) state that for students to learn, they need to receive support from three perspectives: socio-emotional, academic, and linguistic.

However, teaching proactivity is not always in plentiful supply within schools, as evidenced by the Murua-Carton et al. (2012), where it was observed that the lack of preparation on the part of teachers to deal with immigrant students coupled with the lack of willingness of educators means that students do not achieve full integration.

Finally, regarding the third objective, the expectation of socio-educational acceptance of first- and second-generation immigrant students revealed that students belonging to the first generation of immigrants have higher expectations of educational progress than second-generation immigrant students. A similar result was observed in the perception of teaching proactivity for facilitating communication. The data obtained are similar to those of the study by Cebolla (2013), which highlights the so-called paradox of immigrant optimism in which, assuming equal opportunities and conditions among students, immigrant children tend to exhibit greater ambition and expect more results from the educational system compared to their indigenous counterparts.

In the research carried out, it has been shown how first-generation students obtain significant differences in relation to second-generation students in terms of self-reported socio-educational adaptation, perhaps motivated by the fact that they receive greater attention from teachers or have more rapidly adopted the customs and cultures of the host country. This result is in line with those obtained by Rutland et al. (2012), which showed that bi-cultural identifications are related to greater social acceptance and reduced preference for friendships within the same ethnic group. The acquisition of cultural identity in the country of origin and the host country means that these students select multiple group identities, leading to greater social acceptance and neglecting relations with their ethnic groups (Carlson and Güler 2018; Yohani et al. 2019).

## 5. Conclusions

As the main conclusions of this study, schools are undoubtedly a true reflection of the society in which they are contextualized and are increasingly composed of groups with a high degree of internal diversity. Intergroup relationships among young people are increasingly complex and are subject to the diversity of the context. Therefore, to understand the socio-educational adaptation of non-EU students, it is necessary to understand the intergroup dynamics of development and understand that they are all subject to various factors that make them unique.

As Díez Gutiérrez (2020) states in his study on intercultural education, the type of citizenship being constructed significantly influences the development of intercultural projects. It is precisely this influence that advances mutual enrichment, leveraging the diversity present in classrooms and in society to promote all the common elements. This implies a great effort, especially within education policy, as it demands cross-sectional incorporation of multicultural and inclusion themes (Carpani et al. 2011).

The turning point is undoubtedly the element of mutual enrichment since, as has been evidenced in this research, a vast majority of respondents are aware that linguistic mediations would greatly favor their social and school adaptation, making it easier or more bearable. However, it has also been observed that teachers deploy these tools intermittently, highlighting that first-generation immigrant students receive more teaching proactivity. This explains why mastery of the host country's language is scarce or minimal, making it difficult or even impossible to establish social relationships with the rest of the class, even if they are second-generation immigrants.

Therefore, to enrich multicultural environments, it is necessary to use all available resources to facilitate teaching-learning relationships, but above all, to strengthen human relations.

## 6. Ethical Considerations

To this end, specific locations and times were scheduled for the administration of the questionnaire.

Secondly, and following the established planning for data collection, a circular was prepared in which all immigrant students were invited to complete the questionnaires on the day, time and place scheduled.

Third, on the day of data collection, the researchers, vice-dean, teachers, and support teachers were present. At the beginning of the session, students received the necessary instructions to complete the questionnaire, with sufficient time allocated to answer any questions raised by the students. Anonymity and privacy were respected regarding the responses and the students themselves.

**Author Contributions:** Conceptualization, S.C. and J.J.C.-M. methodology, J.J.C.-M. and M.d.M.F.-M.; formal analysis, S.C. and M.d.M.F.-M.; investigation, S.C., J.J.C.-M. and M.d.M.F.-M.; resources, S.C.; data curation, C.M.H.-G. and M.D.P.-E., writing-original draft preparation, M.D.P.-E. and C.M.H.-G.; writing—review and editing, J.J.C.-M. and M.d.M.F.-M.; supervision, J.J.C.-M. and M.d.M.F.-M.; project administration, M.D.P.-E. and C.M.H.-G. All authors have read and agreed to the published version of the manuscript.

**Funding:** This research received no external funding.

**Institutional Review Board Statement:** The study was carried out in accordance with the guidelines of the Code of Good Research Practices of the University of Almería and approved by the University on 7 February 2019.

**Informed Consent Statement:** Informed consent was obtained from all subjects involved in the study.

**Data Availability Statement:** Data supporting the reported results can be found by asking the First author.

**Conflicts of Interest:** The authors declare no conflict of interest.

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
