# Peer review of "The Socio-Educational Adaptation of Secondary School Migrant Students in Sicily: Migrant Generation, School Linguistic Mediation and Teacher Proactivity Factors"

_socsci, doi:10.3390/socsci10070269_

Round 1

Reviewer 1 Report

I think it would be a notable point of improvement for the manuscript to review, firstly, how the objectives are stated and, secondly, to divide them into one or two general objectives and then their corresponding specific objectives. In the same way, the authors should pose some research questions to which they try to provide answers through the study. 

Author Response

Dear Reviewer

The research questions are found in the text of the article between lines 110 and 113. However, we appreciate your observation because it is obvious that they can go unnoticed, so we proceed to give them greater visibility.

Regarding the objectives, we have tried to match their suggestions and that of another reviewer, remaining in a single, very specific hierarchical level and in direct correspondence with the research questions.

Sincerely grateful

Reviewer 2 Report

First of all, I want to thank you for giving me the opportunity to read this paper that in my opinion addresses a relevant topic in the field. However, I have some suggestion to improve the paper, particularly concerning its structure and readability.

  • ABSTRACT:

I would highlight the originality of the work and contribution to previous literature

  • RELATION BETWEEN INTRODUCTION AND METHODOLOGY:

In the final part of section 2, you mention 3 research questions:

1) Do linguistic mediations that occur in school favor these students' integration?

2) Does the educational level of parents influence the socio-educational adaptation of immigrant pupils?

3) Are there differences in socio-educational adaptation between first- and second-generation immigrant students?

In the first part of the methodology, you begin stating that your 3 research questions are:

1) To study the relationships between gender variables, the level of education of the father and mother, and socio-educational adaptation.

2) Analyze the link between linguistic mediations and teaching proactivity concerning educational and social acceptance.

3) Identify the expectation of socio-educational acceptance of first- and second-generation immigrant students.

In my opinion these two formulations may mislead the reader. I think you should maintain the research questions stated in the introduction and link the methodology to them, clearly stating the link between the research questions and the operationalization you have done.

  • METHODOLOGY

Furthermore, you present descriptive statistics concerning dependent variables, but no descriptive statistics concerning independent ones. I think you should enrich the Table 1 and rename it “descriptive statistics” including in it all variables you’ve measured. Maybe I’ll add a correlation table also. Furthermore, you mention you gather 58 items, but you only present a few of them (maybe you can add the interview protocol to facilitate the reader), a reader may be confused about it. I think you should clearly state which data you’ve gathered and why, relating them to previous literature.

Additionally, you should better present in this section how you will link your research questions to data and statistical methods.

  • RESULTS

Again, I think you should present/group results according to the research questions, linking Mann-Whitney U tests in 3 sections, one for each research question, in order to make the paper more readable.

  • DISCUSSION

According to the aforementioned suggestions, I will link results to research questions. For example, you should better state why gender diversity may be something relevant to student integration. Is this a previous literature achievement?

  • CONCLUSIONS

Again, I think the paper need to be improved in its structure. However, conclusions, are better related to the research questions if compared to other sections.

Author Response

Dear reviewer

We appreciate your suggestions, as we understand that they effectively help in a timely manner to improve the quality of the article.

Regarding the linear correspondence between questions and objectives, we understand that their observation facilitates the understanding of the research and we have proceeded to make the suggested modification.

Regarding the linear correspondence between questions and objectives, we understand that their observation facilitates the understanding of the research and we have proceeded to make the suggested modification.

With respect to the observation on descriptions and greater transparency of the survey items, we consider it interesting to indicate that we incorporated the change of name to Table 1. With respect to the inclusion of other variables in the descriptive table, we have incorporated other independent variables. Regarding the number of 58 items, it should be noted that it was a survey that addressed a large number of research questions in the framework of a doctoral thesis, and that we have selected all the items that are contingency related to the sub-object of this article, in order not to saturate the reader with data not relevant to this research sub-object.

The extraction of this subset of item (from the total of the survey) has been thought to include it in a Figure, but finally we attach it to the response file, because we understand that being in Italian, without a simple translation being adequate, the reader could be disoriented.

On the other hand, with respect to the recommendation that it should be presented in this section how the research questions and statistical methods are linked, a new table is created in this regard, a new figure (Figure 2) has bee.

Regarding the Discussion and Conclusions sections, they have focused on the significant results, relatively blurring the linear structure of the research questions.

Finally, grateful and we remain at your disposal

Reviewer 3 Report

Minor spell check required.

Author Response

Dear reviewer

We appreciate your suggestions, as we understand that they effectively help in a timely manner to improve the quality of the article.

Given that the authors do not have a sufficient command of the English language, and we had commissioned the translation to a specialized center, we have asked the translator to review the possible errors that may have been unintentionally produced.

Grateful